# The Importance of Fab Labs in the Development of New Products toward Mass Customization

**Sérgio Carqueijó** [1], **Delfina Ramos** [2], **Joaquim Gonçalves** [1], **Sandro Carvalho** [1,3], **Federica Murmura** [4], **Laura Bravi** [4], **Manuel Doiro** [5], **Gilberto Santos** [6,*] **and Kristína Zgodavová** [7]

1 School of Technology, Polytechnic Institute Cavado Ave, 4750-810 Barcelos, Portugal; sergiocarqueijo96@gmail.com (S.C.); jgoncalves@ipca.pt (J.G.); scarvalho@ipca.pt (S.C.)
2 School of Engineering of Porto (ISEP), Polytechnic of Porto, 4200-465 Porto, Portugal; dgr@isep.ipp.pt
3 2Ai—Applied Artificial Intelligence Laboratory, 4750-810 Barcelos, Portugal
4 Department of Economics, Society, Politics, University of Urbino, 61029 Urbino, Italy; federica.murmura@uniurb.it (F.M.); laura.bravi@uniurb.it (L.B.)
5 Organizacion de Empresas e Márketing, Escola Ingeniería Industrial, Vigo University, 36208 Vigo, Spain; mdoiro@uvigo.es
6 Design School, Polytechnic Institute Cavado Ave, 4750-810 Barcelos, Portugal
7 Faculty of Materials, Metallurgy and Recycling, Institute of Materials and Quality Engineering, Technical University of Košice, 04200 Košice, Slovakia; kristina.zgodavova@tuke.sk
* Correspondence: gsantos@ipca.pt

**Abstract:** The idea of developing Fab Labs (Fabrication Laboratories) was originated by Neil Gershenfeld of the Massachusetts Institute of Technology (MIT) Center for Bits and Atoms in 2006, where it signaled the start of a new era that is changing the world economy by breaking down the boundaries between the digital and physical worlds. The Portuguese Fab Labs are analyzed and a subsequent comparison with others European countries and in the USA was made. This comparison is based on aspects of the profile, the knowledge, the services, and the users. The survey was made by questionnaire, where the Portuguese version of it was adapted from another one disseminated at European level and in the USA, created in connection with a doctoral thesis in Italy. There are 25 active Fab Labs in Portugal, of which 16 responses were obtained and considered valid, so they represent the sample of our study. The results show that the Portuguese Fab Labs are in an embryonic phase with few associated or registered users. Portuguese Fab Labs have areas of work and investment capacity in machinery and technology similar to those of other European countries. However, in terms of turnover, there is a big difference between Portugal and some of the European countries, with American Fab Labs having completely different realities from the European ones. This work is relevant because it compares the Fab Labs of developed countries with those of Portugal. To overcome the difference in good practices existing in other countries, the Portuguese Fab Labs need: (1) better publicity, as well as more support for volunteer workers at Fab Labs, so that more ideas will appear and therefore more products; (2) the facilitation of the use of Fab Labs to have more volunteer workers, who must receive experimental courses, in order to make the best use of the available equipment; (3) evolution, from the current subtractive manufacturing to the additive manufacturing looking for innovation; (4) improved quality, ergonomics, and safety in the design of their own products; (5) and on the part of those responsible, therecognition, dissemination, and celebration of the best ideas that have turned into good products, in order to spread good practices.

**Keywords:** Fab Labs; product development; Mass Customization; Portugal

## 1. Introduction

Over the past few years, the production paradigm has undergone several changes mostly motivated by technological developments [1,2]. The ability to respond to consumers' needs become a fundamental aspect in the competitiveness of companies [3–5]. In this

highly competitive market that requires reduced response times, highly complex products, diversity, and mass production personalization, the reduction of production costs is a problem [6–8]. This requires collaborative spaces, such as Fab labs, which can be defined as a localized space that offers open access to resources, such as machines and prototyping tools [9–11] for industry 4.0 [12–14].

Today, consumers are pushing the technological development of industry, demanding complex, diversified, updated, and even personalized products, as well as very short delivery times. The design and production of "made-to-stock" are changing to "made-to-order"; the Mass Production (MP) is replaced by a production typology based on the need to accommodate new versions and options [15,16]. In various industrial sectors, the globalization of the economy has created a growing need to respond quickly to market demands, which translates into a drastic reduction in the "time to market" new products, making the life cycle of products drastically reduced. The fast and judicious product development becomes a critical factor for the competitiveness and commercial aggressiveness of companies, determining their subsistence capacity [17]. Thus, Mass Customization (MC) began, which is a production system that allows the personification and personalization or individualization of products, as well as services for a value comparable to that of MP [18]. The essence of MC is to transform a customer into a "co-designer", in which the customer is able to gain access and, simultaneously, participate in the design process [15]. The concept of design and product development can be expressed by the requirements or even co-designing the product with the configuration toolkit [19].

Hence, Mass Customization allows a customer to design certain parts or features of a product. With this, the costumer keeps costs closer to that of mass-produced products. In many cases, the components of the product can be modular. This flexibility allows customers to incorporate their ideas in the product [20]. Thus, the customer can mix-and-match options to create a semi-custom final product.

The development process gives rise to essentially interactive and necessarily multidisciplinary activities. These activities allow a large number of methodologies, systems, tools, and solutions developed by professionals and/or companies from different areas, to be shared by all involved. For this integration and unification of views around the product to be developed, the old sheets of paper containing a two-dimensional expression of what was planned to be produced were no longer enough, as they were time-consuming and dubious [21]. One of the most decisive technologies in the renewal of the industry's operation was the introduction of CAD/CAM systems and 3D CAD modelling capabilities. However, although 3D CAD models provide us with a better view of the object under development, they do not offer the tactile sensation or the notion of assembly.

The prototype came to fill this weakness, giving the opportunity to have a better perception of the object under development. According to Jacobs [22], "there is no better way to make sure that a complex piece has all the desired characteristics than to hold it in your hand, rotate it a few times and look at it from all sides", and thus the Rapid Prototyping (RP) and Rapid Tool Manufacturing (RTM) technologies have significantly enhanced the ability to reduce time to market [17]. These prototyping techniques have evolved over the years, and today, they present a much higher execution speed, compared to conventional prototypes. Through the case of the emergence of Fab Labs, a global network of several hundreds of organizations aims to make digital fabrication machines, such as 3D printing, accessible to diverse audiences [23,24]. The transformation of Fab Labs from elite to collective leads many authors to shed light on changes in the governance of innovation processes [25,26].

The aforementioned authors and others focus on the cultural value of making, identifying the methods for sharing the knowledge and the technical skills, particularly, in the context of digital design and fabrication [27–29]. In this case can appear a new trend in engineering education [30–32] able to create new business models with new digital technologies [33–35]. It is very important to create value [36–38] and to protect the intellectual

property [39,40]. At the same time, it is also important to protect the environment [41–43] through concerted actions between organizations [44–46] toward sustainability [47,48].

The activity of producing by adding material (AM) instead of removing it, through 3D printers in which objects are generated by stratification and addition of material, is a revolutionary aspect of prototyping techniques. In addition to the possibility of creating more diversified products with different geometries, the possibility of redefining the activities of the production and logistics processes is offered, new professional figures in the area of manufacturing may be created, new "prototyping platforms for exploration, innovation, invention and learning, providing stimulation for entrepreneurship" [49] as is the case with Fab Labs, which despite developing in a social context, are gradually penetrating the industrial context, in companies such as Airbus, Safran, Airliquide, Orange, and above all at Renault [50,51], which has pioneered the implementation of a corporate Fab Lab [52,53].

The idea of developing Fab Labs (Fabrication Laboratories) was originated by Neil Gershenfeld of the Massachusetts Institute of Technology (MIT) Center for Bits and Atoms in 2006, where it signaled the start of a new era that is changing the world economy by breaking down the boundaries between the digital and physical worlds [49]. The Fab Lab project was created from an experimental course at MIT launched by Gershenfeld in 1998 called "How to Make (Almost) Anything", whose intention was to bring together personal and digital fabrication, individual creativity, and group collaboration. The name illustrates the idea that inspired the Fab Labs: the creation of places where information technology serves the productive activity with a good quality [54–56]. Thus, Fab Labs provide people with the right tools, so they can design and build the most extraordinary things [57], where it exists a bridge from the idea to new product development [58]. In other words, new objects are created with digital design interacting with machines that operate on physical materials [59], where new products are developed [60–62] with designers taking into account the rules of quality [63–65] and also the environmental goal [66–68] of sustainability [69–71]. Yet sometimes problems arise with Indoor Air Quality [72], among others.

The Fab Foundation defines the Manufacture of Laboratories (Fab Lab) as "a technical prototyping platform for innovation and invention, providing a stimulus to local entrepreneurship." At the same time, Fab Lab is a platform for learning and innovation, a place to play, create, learn, guide, invent. A Fab Lab means connection to a global community of students, educators, technologists, researchers, manufacturers, and innovators; in practice, it is a knowledge sharing network that spans 30 countries and 24 time zones. Since all Fab Labs share common tools and processes, the program is building a global network, a distributed laboratory for research and invention [73].

Fab Lab is a prototyping platform for learning and innovation that provides important stimuli for local entrepreneurship and is based mainly on four key factors: openness, interdisciplinary collaboration, effectiveness, and transferability. Currently, the Fab Lab concept is not an alternative to mass production in the creation of large-scale products, but it is committed to demonstrating its potential in modifying the manufacturing logic, offering individuals the ability to create bespoke products, for local and personal needs, to be considered economical according to the logic of mass production [59]. It is a space with a marked social character that offers accessible manufacturing tools and, sometimes, it is conceived as an appropriate platform to quickly start prototyping and development processes of any type of object [74]. In addition, Fab Labs can be incubated by already mature companies, which intend to create laboratories with social, educational, research, and dissemination of their products and services, just like Renault, which is a pioneer in the industrial sector in the development of its own Internal Fab Lab [50]. Increasingly, with respect to their service portfolios, many of them appear to be working likewise to other existing concepts of innovation intermediation such as living laboratories, fab laboratories, business incubators, and co-working spaces [75].

Realizing the impact that the availability and use of Fab Labs can have on the economies of countries, this study aims to understand how Fab Labs are used in Portugal and compares them to similar realities in different countries.

Therefore, the research question that the paper investigates is:

RQ1: What are the differences and similarities among Portuguese Fab Labs and the main European and the American realities of Fab Labs?

The objective of this work was to analyze the use of Fab Labs in Portugal and to compare the sociodemographic and economic reality of Portuguese Fab Labs to the Fab Labs of the main European countries (Italy, France, Germany, Netherlands, and Spain) and the USA.

The structure of this work begins with the introduction, followed by the materials and methods. The results are then presented. Data are analyzed and statistical indicators are displayed. A comparative analysis is made between different countries. Finally, the conclusions appear.

## 2. Material and Methods

A Portuguese version of a questionnaire was developed. It was adapted from another one disseminated at European level and in the USA, created in connection with a doctoral thesis in Italy [59]. There is a total of 25 Fab Labs active in Portugal, found on the Fab Foundation's official website (http://fabfoundation.org/ (accessed on 2 May 2019)); 16 responses were obtained that were considered valid, so they represent the sample of our study. The questionnaire was designed on a digital platform 'Google Forms'. The contacts with the Fab Labs were carried out via telephone, e-mail, or through the Association of Fab Labs Portugal being asked to answer the questionnaire that was available at the indicated link. In all contacts made, it was indicated how the scope of the responses would serve the purpose of the work, the rules of confidentiality and anonymization of data.

In the construction of the questionnaire, three sections were considered:

- The first section investigates the profile of the respective Fab Lab, considering the location, the number of workers, the size of the structure, the revenue, the average number of users, and their investments in machinery and technology;
- The second section describes the skills and competencies of Fab Labs, namely, who are their main customers, what kind of products do they do and what sector are they targeting, what kind of new digital machines do they use most, their main skills, and services that they deliver to customers;
- Section three takes into account the use of digital technologies and the interconnection with the industry. In this section, we try to investigate what percentage of prototypes developed entirely at a Fab Lab actually arrived in the industry, also to understand the level of connection of Fab Labs with the prototyping industries and beyond, as well as to understand if external organizations choose to develop projects to address deficiencies in the Fab Labs incubators. In this part, the contribution of Quality Management Systems (QMSs), namely through metrology, to product innovation within the scope of Fab Labs will also be investigated. At the end of the section, it was also asked, based on experiences, customers' demand for new projects and the experience of the market's evolution and progress, which technologies will be decisive in the near future.

The first contacts took place in July 2019, and the data collection process was completed in September. The data were processed and analyzed with the statistical software SPSS (IBM SPSS Statistics 22, Armonk, NY, USA) and the graphics created with the Microsoft Office Excel.

## 3. Results

### 3.1. Data Analysis

In order to study the different competence dimensions of the Fab Labs, we carried out a Principal Component Analysis (PCA), with Oblimin rotation [76,77].

Variables whose factor loadings are less than 0.6 were excluded from the analysis. The scale used for the variables under study was a 5-point Likert scale. All tests were performed with 95% confidence. In all next tables, N represents the number of Fab Labs surveyed, which is 16.

### 3.2. Statistical Indicators

A descriptive analysis was carried out to highlight the main characteristics of Fab Labs in order to determine their socio-demographic and economic profile. Tables 1–7 show these indicators. In addition, Cronbach's alpha value was measured to evaluate discriminating capacity of each questions group [78]. It was considered that any group with an alpha value greater than or equal to 0.6 have a discriminating capacity. Table 1 shows who are the main Portuguese Fab Labs users. The individual customers (3.44) are the customers with which the laboratories work most. On the other hand, it is manufacturing companies (2.06) that least seek the services offered by Fab Labs.

**Table 1.** Main Portuguese Fab Labs users ($\alpha$ = 0.862).

|  | N | Minimum | Maximum | Average | Standard Deviation |
|---|---|---|---|---|---|
| Manufacturing companies | 16 | 1 | 4 | 2.06 | 0.854 |
| Individual customers | 16 | 2 | 5 | 3.44 | 1.365 |
| Professionals | 16 | 2 | 5 | 3.00 | 1.033 |
| Institutions/Schools | 16 | 1 | 4 | 2.81 | 1.109 |
| Universities | 16 | 1 | 5 | 2.38 | 1.088 |
| Artists | 16 | 1 | 5 | 2.81 | 1.167 |
| Designers | 16 | 1 | 5 | 2.81 | 1.167 |

**Table 2.** Sectors with presence in Portuguese FabLabs ($\alpha$ = 0.523).

|  | N | Minimum | Maximum | Average | Standard Deviation |
|---|---|---|---|---|---|
| Fashion | 16 | 1 | 4 | 2.38 | 1.025 |
| Wood Industry | 16 | 1 | 5 | 2.56 | 1.209 |
| Mechanic | 16 | 1 | 4 | 2.25 | 0.931 |
| Automotive | 16 | 1 | 3 | 1.50 | 0.730 |
| Food | 16 | 1 | 4 | 1.94 | 0.998 |
| Electronic Technology | 16 | 1 | 4 | 2.56 | 0.814 |
| IoT | 16 | 1 | 4 | 2.31 | 0.946 |
| Software | 15 | 1 | 5 | 1.93 | 1.100 |

**Table 3.** Developed product type ($\alpha$ = 0.770).

|  | N | Minimum | Maximum | Average | Standard Deviation |
|---|---|---|---|---|---|
| Products—commercialization | 16 | 1 | 4 | 2.31 | 0.946 |
| Products—single customer | 16 | 2 | 4 | 2.75 | 0.856 |
| Prototypes—enterprises | 15 | 1 | 3 | 2.07 | 0.594 |
| Prototypes—single customer | 16 | 1 | 5 | 2.25 | 0.856 |

**Table 4.** Frequency of use of equipment in Portuguese Fab Labs ($\alpha$ = 0.453).

|  | N | Minimum | Maximum | Average | Standard Deviation |
|---|---|---|---|---|---|
| 3D Printer | 16 | 2 | 5 | 3.69 | 1.014 |
| 3D Scanner | 16 | 1 | 6 | 3.00 | 1.897 |
| Laser cutting machine | 16 | 2 | 6 | 4.63 | 0.885 |
| CNC Milling machine | 16 | 2 | 6 | 4.31 | 1.014 |
| Vinyl cutter | 16 | 2 | 6 | 3.37 | 1.310 |
| Lathe | 16 | 1 | 6 | 3.25 | 1.880 |
| Quality control charts | 16 | 2 | 6 | 2.69 | 1.250 |
| Precision punch | 16 | 1 | 6 | 3.56 | 2.308 |

**Table 5.** Frequency of services provided by Portuguese Fab Labs ($\alpha = 0.834$).

|  | N | Minimum | Maximum | Average | Standard Deviation |
|---|---|---|---|---|---|
| Product printing | 16 | 2 | 5 | 3.62 | 1.088 |
| Prototypes creation support | 16 | 2 | 5 | 3.13 | 1.147 |
| New product design/support | 16 | 1 | 5 | 3.25 | 1.238 |
| Support to redefinition of production process | 16 | 1 | 5 | 2.50 | 1.317 |
| Materials consulting | 16 | 1 | 4 | 2.25 | 0.931 |
| Experimental courses | 16 | 1 | 4 | 2.25 | 1.065 |

**Table 6.** Competences of Portuguese Fab Labs ($\alpha = 0.641$).

|  | N | Minimum | Maximum | Average | Standard Deviation |
|---|---|---|---|---|---|
| Arduino programming | 16 | 2 | 5 | 3.63 | 1.088 |
| Software programming | 16 | 1 | 5 | 3.00 | 1.211 |
| Design software | 16 | 3 | 5 | 4.31 | 0.793 |
| Hardware | 16 | 1 | 5 | 3.56 | 1.209 |
| Materials | 16 | 2 | 5 | 4.25 | 0.775 |
| Business process | 16 | 2 | 5 | 3.62 | 1.147 |
| IoT | 16 | 1 | 5 | 3.25 | 1.065 |
| Digital manufacturing | 15 | 3 | 5 | 4.47 | 0.640 |

**Table 7.** Factors considered in products design ($\alpha = 0.500$).

|  | N | Yes | No |
|---|---|---|---|
| Design | 16 | 14 | 2 |
| Quality | 16 | 11 | 5 |
| Ergonomics | 16 | 5 | 11 |
| Security | 16 | 9 | 7 |
| Ecology | 16 | 13 | 3 |

Table 2 presents data about the main industrial sectors that turn to Fab Labs. The wood industry sector, and electronic technology sector (2.56) are more present in Portuguese Fab Labs. The automotive sector (1.50) is the sector that least seeks the services offered by Portuguese Fab Labs.

Table 3 shows the number of types of products made. It appears that products made for a single customer are the majority, and these data are in line with those obtained in Table 1 where we analyze that in Portugal it is the individual customers that most attend Fab Labs. On the other hand, prototypes for companies are those that are made in smaller quantities by Fab Labs, again in line with data in Table 1, where it was possible to observe that manufacturing companies were the ones that least used the services of Portuguese Fab Labs.

Table 4 shows that, despite 3D printer being the 3rd most used equipment in Fab Labs, the most used equipment in Portuguese Fab Labs are, respectively, laser cutting machines and CNC milling machines, equipment that still belongs to subtractive manufacturing. This proves the reality of Portuguese companies that have not yet taken the leap towards a more ecological and sustainable manufacture as the additive manufacturing. The least used devices are the quality control charts, followed by the 3D scanner.

Data on Table 5 allow us to conclude that the service most frequently provided in Portuguese Fab Labs is the printing of products, proving the idea that 3D printers are one of the most used equipment in Portuguese Fab Labs. With the same value, the two services provided less frequently are materials consulting and experimental courses.

Regarding the Portuguese Fab Labs capabilities (Table 6), it is possible to observe that they are more oriented to the digital manufacturing area, followed by design materials

and software. On the other hand, they have less capabilities in the field related with software programming.

Analyzing which factors are most considered in the design of products in the Fab Labs (Table 7), the most Fab Labs give great importance, in this order, on design, eco sustainability and product quality.

With regard to security, this field is balanced between those who give more importance to security and those who give less importance. However, there is a tendency towards those who place more importance on security.

Conversely, in terms of product ergonomics, there are still few who consider this field in the design of their products.

### 3.3. Main Component Analysis

For a better understanding of the structure of the different sectors in the Fab Labs, a Principal Analysis Components (PCA) was realized, for each study element, which is discussed below (Table 8).

**Table 8.** PCA—Main users of Portuguese Fab Labs ($\alpha$ = 0.862).

| | Components | |
| --- | --- | --- |
| | **Concretization of Ideas** | **Research** |
| Manufacturing companies | - | 0.896 |
| Individual customers | 0.740 | - |
| Professionals | 0.906 | - |
| Institutions/Schools | 0.714 | - |
| Universities | - | 0.927 |
| Artists | 0.955 | - |
| Designers | 0.598 | - |
| KMO | 0.645 | |
| % Cumulative Variance | 55.087 | 74.672 |

Extraction Method: Principal Component Analysis. Rotation Method: Oblimin with Kaiser Normalization. Converged rotation in 5 iterations.

After conducting PCA related to the main sectors with which the Portuguese Fab Labs work (Table 9), the existence of 3 components can be observed. The KMO value is 0.481, which indicates that the proportion of the variance that the variables have in common is quite low.

**Table 9.** PCA—Sectors with which the Portuguese Fab Labs work ($\alpha$ = 0.523).

| | Components | | |
| --- | --- | --- | --- |
| | **1** | **2** | **3** |
| Fashion | - | 0.784 | - |
| Wood Industry | - | 0.938 | - |
| Mechanic | 0.941 | - | - |
| Automotive | - | - | 0.732 |
| Food | - | - | 0.950 |
| Electronic Technology | 0.925 | - | - |
| IoT | 0.876 | - | - |
| Software | - | 0.655 | - |
| KMO | | 0.481 | |
| % Cumulative Variance | 38.915 | 61.171 | 77.060 |

Extraction Method: Principal Component Analysis. Rotation Method: Oblimin with Kaiser Normalization. Converged rotation in 5 iterations.

After conducting a PCA applied to the type of products performed by the Portuguese Fab Labs (Table 10), we found that it was possible to group all types of products into just

one component, called Product Development. The KMO value is 0.649, which indicates that the proportion of the variance that the variables have in common is reasonable.

**Table 10.** PCA—Type of products made ($\alpha$ = 0.770).

| | Component Product Development |
|---|---|
| Products—commercialization | 0.721 |
| Product—single customer | 0.703 |
| Prototypes—enterprises | 0.797 |
| Prototypes—single customer | 0.888 |
| KMO | 0.649 |
| % Cumulative Variance | 60.960 |

Extraction Method: Principal Component Analysis. 1 component extracted.

After executing a PCA on the frequency of use of the equipment in the Portuguese Fab Labs (Table 11), we found that it was possible to group all the equipment into 3 distinct components. The KMO value is 0.404, which indicates that the proportion of variance that the variables have in common is quite low.

**Table 11.** PCA—Frequency of use of equipment in Portuguese Fab Labs ($\alpha$ = 0.453).

| | Component | | |
|---|---|---|---|
| | 1 | 2 | 3 |
| 3D Printer | - | - | 0.834 |
| 3D Scanner | 0.687 | - | - |
| Laser cutting machine | - | - | 0.604 |
| CNC Milling machine | 0.838 | - | - |
| Vinyl cutter | 0.654 | - | - |
| Lathe | - | 0.650 | - |
| Quality control charts | - | - | 0.680 |
| Precision punch | - | 0.900 | - |
| KMO | | 0.404 | |
| % Cumulative Variance | 27.869 | 50.075 | 66.411 |

Extraction Method: Principal Component Analysis. Rotation Method: Oblimin with Kaiser Normalization. Converged rotation in 10 iterations.

After a PCA was carried out on the frequency of provision of services offered by the Portuguese Fab Labs (Table 12), it was verified that it was possible to group all types of products in just one component, called Support, training and product realization. Support and training is important, because people need to know how to work with machines to make their own product of their own design from a good idea. The KMO value is 0.727, which indicates that the proportion of the variance that the variables have in common is reasonable.

**Table 12.** PCA—Frequency of the provision of services offered by the Fab Labs ($\alpha$ = 0.834).

| | Component Support. Training and Product Realization |
|---|---|
| Product printing | 0.606 |
| Prototypes creation support | 0.807 |
| New product design/support | 0.700 |
| Support to redefinition of production process | 0.793 |
| Materials consulting | 0.780 |
| Experimental courses | 0.767 |
| KMO | 0.727 |
| % Cumulative Variance | 55.545 |

Extraction Method: Principal Component Analysis. 1 component extracted.

Related with skills of the Portuguese Fab Labs (Table 13) the PCA reveals that it was possible to group all equipment into 3 distinct components, the Skills in Hardware, Programming and Business (cumulative variance of 41.20%), Skills in the creation of ideas (cumulative variance of 63.08%), and Skills in materials (cumulative variance of 79.25%). The KMO value is 0.505, which indicates that the proportion of variance that the variables have in common is low.

**Table 13.** PCA—Competences of Portuguese Fab Labs ($\alpha$ = 0.641).

| | Component—Skill | | |
| | Hardware. Programming and Business | Idea Creation | Materials |
|---|:---:|:---:|:---:|
| Arduino programming | 0.683 | - | - |
| Software programming | 0.724 | - | - |
| Design software | - | 0.841 | - |
| Hardware | 0.839 | - | - |
| Materials | - | - | 0.984 |
| Business process | 0.672 | - | - |
| IoT | 0.932 | - | - |
| Digital manufacturing | - | 0.879 | - |
| KMO | | 0.505 | |
| % Cumulative Variance | 41.200 | 63.079 | 79.251 |

Extraction Method: Principal Component Analysis. Rotation Method: Oblimin with Kaiser Normalization. Converged rotation in 7 iterations.

By developing a PCA on the factors considered in the product design by the Portuguese Fab Labs (Table 14), we realize the existence of two components. The first group was called Quality in well-being (cumulative variance of 35.91%) and groups the type of quality tangible by the user in the use and handling of the product. The second group, called Indirect Quality (cumulative variance of 64.37%), is the group consisting of product quality other than ease of handling, that is, the quality of the design and the ecological quality of the products. The KMO value is 0.448, which indicates that the proportion of variance that the variables have in common is quite low.

**Table 14.** PCA—Factors considered when designing products ($\alpha$ = 0.500).

| | Components—Quality | |
| | Well-Being Quality | Indirect Quality |
|---|:---:|:---:|
| Design | - | 0.820 |
| Quality | 0.578 | - |
| Ergonomics | 0.786 | - |
| Security | 0.852 | - |
| Ecology | - | 0.700 |
| KMO | | 0.448 |
| % Cumulative Variance | 35.913 | 64.369 |

Extraction Method: Principal Component Analysis. Rotation Method: Oblimin with Kaiser Normalization. Converged rotation in 9 iterations.

### 3.4. Comparative Analysis of Results among Countries

In order to ascertain the existence of statistically significant differences between countries in relation to the factors analyzed, a Kruskal-Wallis test (for each variable) was performed, which is the non-parametric test corresponding to the ANOVA (Analyzes of Variance) parametric test that it was not used because the sample was small (N = 16 < 30). The test was carried out with 90% confidence, that is, the null hypothesis will be rejected in cases where the proof value ("sig." in SPSS) is less than 10%. At the same time, Bonferroni's post-hoc comparison tests were performed [79]. In the Post-hoc tests, we only present the lines where there were statistically significant differences from Portugal.

Table 15 shows the countries of all Fab Labs that entered the study. In a very succinct way, we can observe that Portugal is the country with the lowest number of Fab Labs spread throughout its territory, despite being the country with the highest response rate to the questionnaire.

**Table 15.** List of Fab Labs present in each country and those contacted.

|  | Portugal | Italy | France | Germany | Netherland | Spain | USA | Total |
|---|---|---|---|---|---|---|---|---|
| Fab Labs in country | 25 | 134 | 151 | 46 | 32 | 46 | 158 | 592 |
| Contacted Fab Labs | 25 | 112 | 142 | 41 | 29 | 42 | 127 | 518 |
| Responses | 16 | 27 | 16 | 5 | 3 | 8 | 14 | 89 |
| % of responses | 64.00 | 24.11 | 11.27 | 12.20 | 10.34 | 19.05 | 11.02 | 17.18 |

Table 16 allows the analysis of a set of features of the generic profile of the Fab Labs for each country. Thus, in a first approach, it is possible to observe that in only 3 of the 7 countries under study there are more than 20 volunteers in at least one Fab Lab, Portugal being one of those that does not have a number of volunteers above that value. Regarding the Fab Labs' work area, we can see that Portugal has good work areas in its Fab Labs, compared to the other countries and is also well ranked in terms of the number of associated or registered users. However, considering the economic level, it is possible to detect that the Portuguese Fab Labs earn an annual revenue well below the European average, which is why it is the country with the lowest revenue both at European level and in comparison with the USA. It can be also observed that the average revenue in the USA is higher than the sum of the revenues of the European countries under study. With regard to investment in technology and machinery, Portugal is relatively well matched with the other European countries under study, with one of its Fab Labs having an investment between 300,000–500,000 €, whereas at European level only Germany has a Fab Labs with a higher investment. Comparing Europe with the USA, it turns out, once again, that the USA is quite distant from Europe, with two of its Fab Labs with investments exceeding €1,000,000. Finally, we find that Portugal is the country where Fab Labs receive the major number of state or European incentives, with 73.3% of Fab Labs receiving some type of these incentives.

**Table 16.** Fab Labs profile in Europe and USA.

|  | Portugal (16) | | Italy (27) | | France (16) | | Germany (5) | | Netherlands (3) | | Spain (8) | | USA (14) | |
|---|---|---|---|---|---|---|---|---|---|---|---|---|---|---|
|  | N | % | N | % | N | % | N | % | N | % | N | % | N | % |
| *Volunteer Workers* | | | | | | | | | | | | | | |
| <20 | 16 | 100 | 17 | 100 | 11 | 68.7 | 3 | 60.0 | 3 | 100 | 8 | 100 | 12 | 85.7 |
| >20 | 0 | 0.0 | 0 | 0.0 | 5 | 31.3 | 2 | 40.0 | 0 | 0.0 | 0 | 0.0 | 2 | 14.3 |
| *Fab Labs dimension (square meters)* | | | | | | | | | | | | | | |
| 5–24 | 0 | 0.0 | 1 | 3.7 | 1 | 6.3 | 0 | 0.0 | 0 | 0.0 | 1 | 12.5 | 3 | 21.4 |
| 25–74 | 4 | 25.0 | 9 | 33.3 | 8 | 50.0 | 2 | 40.0 | 0 | 0.0 | 1 | 12.5 | 2 | 14.3 |
| 75–200 | 8 | 50.0 | 12 | 44.4 | 5 | 31.3 | 2 | 40.0 | 3 | 100 | 5 | 62.5 | 4 | 28.6 |
| >200 | 4 | 25.0 | 5 | 18.5 | 2 | 12.5 | 1 | 20.0 | 0 | 0.0 | 1 | 12.5 | 5 | 35.7 |
| *Registry or associate users* | | | | | | | | | | | | | | |
| <50 | 9 | 59.3 | 16 | 59.2 | 7 | 53.8 | 2 | 40.0 | 1 | 33.3 | 6 | 75.0 | 5 | 35.7 |
| 50–100 | 2 | 12.5 | 5 | 18.5 | 5 | 31.3 | 0 | 0.0 | 0 | 0.0 | 0 | 0.0 | 1 | 7.1 |
| >100 | 5 | 31.3 | 6 | 22.2 | 4 | 25.0 | 3 | 60.0 | 2 | 66.7 | 2 | 25.0 | 8 | 57.1 |
| Fab Labs annual revenue | 8.59 € | | 31.88 € | | 34.35 € | | 10.50 € | | 35.50 € | | 15.33 € | | 154.285 $ | |

**Table 16.** *Cont.*

| | Portugal (16) | | Italy (27) | | France (16) | | Germany (5) | | Netherlands (3) | | Spain (8) | | USA (14) | |
|---|---|---|---|---|---|---|---|---|---|---|---|---|---|---|
| | N | % | N | % | N | % | N | % | N | % | N | % | N | % |
| Investment in technology and machinery (in thousands of euros) | | | | | | | | | | | | | | |
| <10 | 7 | 43.8 | 14 | 51.9 | 9 | 56.3 | 1 | 20.0 | 2 | 66.7 | 4 | 50.0 | 0 | 0.0 |
| 10–50 | 4 | 25.0 | 7 | 25.9 | 5 | 31.3 | 2 | 40.0 | 1 | 33.3 | 3 | 37.5 | 3 | 21.4 |
| 50–100 | 3 | 18.8 | 4 | 14.8 | 2 | 12.5 | 1 | 20.0 | 0 | 0.0 | 0 | 0.0 | 5 | 35.7 |
| 100–300 | 1 | 6.3 | 1 | 3.7 | 0 | 0.0 | 0 | 0.0 | 0 | 0.0 | 1 | 12.5 | 2 | 14.3 |
| 300–500 | 1 | 6.3 | 1 | 3.7 | 0 | 0.0 | 0 | 0.0 | 0 | 0.0 | 0 | 0.0 | 1 | 7.1 |
| 500–1000 | 0 | 0.0 | 0 | 0.0 | 0 | 0.0 | 1 | 20.0 | 0 | 0.0 | 0 | 0.0 | 1 | 7.1 |
| >1000 | 0 | 0.0 | 0 | 0.0 | 0 | 0.0 | 0 | 0.0 | 0 | 0.0 | 0 | 0.0 | 2 | 14.3 |
| State or European funds | | | | | | | | | | | | | | |
| Yes | 11 | 73.3 | 11 | 40.7 | 6 | 37.5 | 2 | 40.0 | 1 | 33.3 | 1 | 12.5 | 6 | 42.9 |
| No | 4 | 26.7 | 16 | 59.3 | 10 | 62.5 | 3 | 60.0 | 2 | 66.7 | 7 | 87.5 | 8 | 57.1 |

A KrusKal-Wallis test was carried out to ascertain the possible existence of differences between different countries in relation to the main users of Fab Labs (Table 17). The test results show that there are some differences between countries. The differences are found in manufacturing companies, institutions/schools, artists, and designers, while for individual customers, professionals, and universities, no significant differences were found.

**Table 17.** Kruskal-Wallis test for Main users of Fab Labs analysis between countries.

| Main Users | Sig. |
|---|---|
| Manufacturing companies | 0.071 |
| Individual customers | 0.107 |
| Professionals | 0.521 |
| Institutions/Schools | 0.029 |
| Universities | 0.150 |
| Artists | 0.076 |
| Designers | 0.071 |

The same test was used to analyze differences between the sectors with which Fab Labs work (Table 18). It is possible to conclude that some evident differences between Fab Labs from different countries exist. These evidences were detected with respect to the fashion, mechanics, food, electronic technology, and IoT sectors. In contrast, in the wood industry sector, in the automotive sector and in software technology, no differences were found between Fab Labs of the various countries under study.

**Table 18.** Kruskal-Wallis—Sectors that Fab Labs work with.

| Sectors | Sig. |
|---|---|
| Fashion | 0.002 |
| Wood Industry | 0.421 |
| Mechanic | 0.026 |
| Automotive | 0.444 |
| Food | 0.016 |
| Electronic Technology | 0.009 |
| IoT | 0.019 |
| Software | 0.251 |

Subsequently, a Post-hoc test was performed, in order to detect where differences exist between different countries. However, only the differences in which Portuguese Fab Labs

are involved were selected. Thus, it can be seen that, according to the responses to the questionnaires, there are differences between Portugal and Germany in the mechanical sector, as well as, in the electronic technology and IoT. In the IoT sector, Portugal is also different from France. It should be noted that in all these comparisons, Portugal is always at a lower level than other countries, as we can see in the column of the 90% confidence interval, in which both limits are negative. In addition to the aforementioned differences, there is a tendency for other differences to be seen, even though these are not so evident and therefore were not considered (the 90% confidence interval contains 0 between the lower and upper limit). To confirm the trend previously analyzed, Portugal in the electronic technology sector is also tending to be at a lower level compared to France and Italy. Moreover, it can be considered the case of the fashion sector, where Portugal in comparison with neighboring Spain is at a higher level, that is, the Portuguese Fab Labs work more for the fashion sector than the Spanish Fab Labs. These results can be observed in Table 19.

**Table 19.** Post-hoc Test—Sectors that Fab Labs work with.

| Sectors | Country | Sig. | Lower Limit | Upper Limit |
|---|---|---|---|---|
| Fashion | Spain | 0.133 | −0.04 | 2.29 |
| Mechanic | Germany | 0.03 | −3.29 | −0.21 |
| Electronic Technology | France | 0.13 | −2.16 | 0.03 |
| | Germany | 0.008 | −3.63 | −0.45 |
| | Italy | 0.165 | −1.9 | 0.06 |
| IoT | France | 0.047 | −2.4 | −0.1 |
| | Germany | 0.089 | −3.35 | −0.02 |

A Kruskal-Wallis test was performed to determine differences between countries considering the types of products performed in Fab Labs (Table 20). The test indicates that there is a difference only in the prototypes for a single customer, and in the remaining hypotheses, no differences were found between the various countries.

**Table 20.** Kruskal-Wallis test—Type of products produced.

| Products Type | Sig. |
|---|---|
| Products—commercialization | 0.271 |
| Products—single customer | 0.112 |
| Prototypes—enterprises | 0.397 |
| Prototypes—single customer | 0.025 |

The difference found in Table 20 is in line with the difference found between Portugal and Italy, after the Post-hoc test was realized (Table 21). This difference confirms that Portuguese Fab Labs manufacture fewer prototypes for a single customer than Italian Fab Labs.

**Table 21.** Post-hoc Test—Type of products performed.

| Prototypes | Country | Sig. | Lower Limit | Upper Limit |
|---|---|---|---|---|
| Prototypes for a single costumer | Italy | 0.041 | −1.92 | −0.09 |

Using the Kruskal-Wallis test yielded the results that are shown in Table 22; differences were detected between countries considering the main equipment they use in Fab Labs, namely in CNC milling machines, vinyl cutters, lathe, and also in precision punching for printed circuits, while in the rest hypotheses no differences were found between different countries.

**Table 22.** Kruskal-Wallis test—Frequency of use of equipment in Fab Labs.

| Frequency of Use of Equipment | Sig. |
|---|---|
| 3D Printer | 0.215 |
| 3D Scanner | 0.816 |
| Laser cutting machine | 0.282 |
| CNC Milling machine | 0.019 |
| Vinyl cutter | 0.019 |
| Lathe | 0.004 |
| Quality control charts | 0.083 |
| Precision punch | 0.092 |

A Post-hoc test, presented in Table 23, was carried out, where it is possible to detect if there are differences between Portugal and the other countries. As it can be seen, Portugal is different from other countries in the use of CNC milling machines, lathe and precision punching for printed circuits. All of these differences show that Portugal uses this equipment more in its Fab Labs. This can be a negative indicator for Portugal, since both CNC milling machines and the lathe are subtractive manufacturing equipment, and the future depends on the use of machinery where additive technology is predominant.

**Table 23.** Post-hoc test—Frequency of use of equipment.

| Frequency of Use of Equipment | Country | Sig. | Lower Limit | Upper Limit |
|---|---|---|---|---|
| Milling machines CNC | France | 0.013 | 0.28 | 2.72 |
| Lathe | Netherlands | 0.059 | 0.14 | 4.49 |
| | Germany | 0.055 | 0.09 | 2.66 |
| | Spain | 0.076 | 0.05 | 3.2 |
| | USA | 0.009 | 0.35 | 3.01 |
| | Italy | 0.001 | 0.55 | 2.84 |
| Precision punch | France | 0.043 | 0.15 | 3.22 |
| | USA | 0.024 | 0.26 | 3.44 |

As for the frequency of provision of services offered by Fab Labs, the Kruskal-Wallis test performed (Table 24) allows us to conclude that there are differences between countries regarding materials consultancy and experimental courses, and in the remaining hypotheses, no differences were found between countries.

**Table 24.** Kruskal-Wallis—Frequency of provision of services offered by Fab Labs.

| Frequency of Provision of Services | Sig. |
|---|---|
| Product printing | 0.750 |
| Prototypes creation support | 0.144 |
| New product design/support | 0.331 |
| Support to redefinition of production process | 0.558 |
| Materials consulting | 0.035 |
| Experimental courses | 0.000 |

Once again, the differences found in the frequency of service provision are in line with the differences found in Portugal, after the Post-hoc test, is performed (see Table 25). These differences show us that Portugal, in relation to experimental courses, is unlike everyone else. This can also be a bad indicator for Portuguese Fab Labs, since Portugal provides less experimental course services. In addition to the mentioned differences, there are other differences, even though these are not so evident. Portuguese Fab Labs regarding materials consultancy are also tending to be at a lower level, that is, Portuguese Fab Labs

tend to provide less services in this field, compared to Germany and USA. Possibly, this can be attributed to Portugal having a low conception of products and its industry being traditionally one of production instead of conception of goods.

**Table 25.** Post-hoc Test—Frequency of services provided by Fab Labs.

| Frequency of Services Provided | Country | Sig. | Lower Limit | Upper Limit |
|---|---|---|---|---|
| Materials Consultancy | Germany | 0.131 | −3.56 | 0.06 |
| | USA | 0.131 | −2.54 | 0.04 |
| Experimental Courses | France | 0.003 | −2.28 | −0.35 |
| | Netherlands | 0.085 | −3.47 | −0.03 |
| | Germany | 0 | −3.75 | −0.95 |
| | Spain | 0 | −3.68 | −1.32 |
| | USA | 0 | −3.32 | −1.32 |
| | Italy | 0 | −2.46 | −0.74 |

The KrusKal-Wallis test, carried out in relation to the skills of Fab Labs (Table 26), shows differences, between countries in skills in materials, business processes, IoT, and digital manufacturing, and in the remaining hypotheses, no differences were found between the various countries.

**Table 26.** Kruskal-Wallis test—Fab Labs skills.

| Fab Labs Skills | Sig. |
|---|---|
| Arduino programming | 0.673 |
| Software programming | 0.210 |
| Design software | 0.433 |
| Hardware | 0.189 |
| Materials | 0.097 |
| Business process | 0.072 |
| IoT | 0.059 |
| Digital manufacturing | 0.345 |

Table 27 shows the Post-hoc test, which allows to observe differences between Portugal and France in terms of skills in materials and business processes. However, the focus is on differences in IoT skills. It is visible that Portugal is different from all the others in a negative way, that is, the Portuguese Fab Labs have less skills in IoT than all the others. Since IoT, today, is one of the most innovative technologies and with a very high progression margin in the near future, this is certainly a bad indicator for Portuguese Fab Labs.

**Table 27.** Post-Hoc Test—Fab Labs skills.

| Skills | Country | Sig. | Lower Limit | Upper Limit |
|---|---|---|---|---|
| Materials | France | 0.097 | 0 | 1.87 |
| Business Process | France | 0.072 | 0.04 | 2.21 |
| IoT | France | 0 | −3.69 | −1.44 |
| | Netherlands | 0.059 | −4.13 | −0.12 |
| | Germany | 0 | −4.15 | −0.9 |
| | Spain | 0 | −4 | −1.25 |
| | USA | 0 | −3.07 | −0.75 |
| | Italy | 0 | −3.39 | −1.38 |

The Kruskal–Wallis test by ranks is a non-parametric method for testing whether samples originate from the same distribution. It is used for comparing two or more independent samples of equal or different sample sizes. A significant Kruskal–Wallis test

indicates if at least one sample stochastically dominates another sample. The Kruskal-Wallis test performed in relation to the factors considered in the design of products (Table 28) shows the existence of differences, if they exist, in all the factors considered.

**Table 28.** Kruskal-Wallis test—Factors considered when designing products.

| Factors to Designing Products | Sig. |
|---|---|
| Design | 0.000 |
| Product quality | 0.000 |
| Ergonomics | 0.000 |
| Security | 0.000 |

The differences found in Table 28 confirm the differences found with Portugal. The Post-hoc test (Table 29) show that Portugal is different from all the others in all the factors considered. This is also a bad indicator for Portuguese Fab Labs, since Portuguese Fab Labs are the ones that least consider the different factors compared to all the others.

**Table 29.** Post-hoc Test—Factors considered in product design.

| Factors | Country | Sig. | Lower Limit | Upper Limit |
|---|---|---|---|---|
| Design | France | 0 | −3.02 | −1.23 |
| | Netherlands | 0 | −4.38 | −1.2 |
| | Germany | 0 | −4.02 | −1.43 |
| | Spain | 0 | −3.97 | −1.78 |
| | USA | 0 | −4.26 | −2.41 |
| | Italy | 0 | −3.81 | −2.22 |
| Product Quality | France | 0 | −3.06 | −1.06 |
| | Netherlands | 0.001 | −4.43 | −0.86 |
| | Germany | 0 | −4.36 | −1.46 |
| | Spain | 0 | −3.79 | −1.34 |
| | USA | 0 | −3.99 | −1.92 |
| | Italy | 0 | −3.95 | −2.16 |
| Ergonomics | France | 0 | −3.6 | −1.52 |
| | Netherlands | 0.009 | −4.21 | −0.5 |
| | Germany | 0 | −4.6 | −1.58 |
| | Spain | 0 | −3.96 | −1.41 |
| | USA | 0 | −3.98 | −1.82 |
| | Italy | 0 | −4.36 | −2.5 |
| Safety | France | 0 | −3.17 | −1.2 |
| | Netherlands | 0.09 | −3.52 | −0.02 |
| | Germany | 0 | −4.26 | −1.41 |
| | Spain | 0 | −3.52 | −1.11 |
| | USA | 0 | −2.95 | −0.92 |
| | Italy | 0 | −3.5 | −1.75 |

## 4. Conclusions

Fab Labs are known for being small workshops where anyone, institution, or company can develop or create something new. There are places where it is possible to do things, but there are other ones where it is difficult to do something, so Fab Labs offer a variety of very versatile equipment and a diverse range of services. When the sociodemographic and economic reality of Portuguese Fab Labs is compared with the Fab Labs of the main European countries (Italy, France, Germany, Netherlands, and Spain) and the USA, the results obtained show that, in the Portuguese reality, there are still some Fab Labs in an embryonic phase with few associated or registered users, but, on the other hand, others already have another maturity with more than 100 users. The number of volunteer workers also demonstrates that the Portuguese Fab Labs are not yet in the size of some of the Fab

Labs in other countries. Portuguese Fab Labs have areas of work and investment capacity in machinery and technology similar to those of other European countries. However, in terms of turnover, there is a big difference between Portugal and some of the other European countries, with American Fab Labs having completely different realities from the European ones, with a turnover of more than 6 times compared to the European average.

There are also many differences regarding experimental courses between Portugal and the other countries, and this indicator may be a barrier to innovation, information, and knowledge of new technologies. This indicator may be related to other results obtained, namely with the fact that Portuguese Fab Labs have less consideration for factors such as quality, ergonomics, safety in the design of their own products, which may be caused by a lack of knowledge. The Fab Labs should focus on ideas that can be transformed into new products. Hence, ideas capable of being turned into products are needed. Knowing what other countries are doing will help those who are further behind.

This work is relevant because it compares the FAB Labs of developed countries with those of Portugal. To overcome the difference in good practices existing in other countries, the Portuguese Fab labs need: (1) better publicity, more support for volunteer workers at FAB Labs, so that more ideas will appear and therefore more products; (2) the facilitation of the use of FAB Labs to have more volunteer workers, who must receive experimental courses, in order to make the best use of the available equipment; (3) evolution, from the current subtractive manufacturing to the additive manufacturing looking for innovation; (4) improved quality, ergonomics, and safety in the design of their own products; (5) and on the part of those responsible, the recognition, dissemination, and celebration of the best ideas that have turned into good products, in order to spread good practices.

However, it is pertinent to highlight the existence of some limitations in the research. In fact, the existence of Fab Labs in Portugal is still very small, being, compared to the other countries considered in the study, the country with the least number of Fab Labs. However, Portugal is also the country with the lowest population among the countries under study and the 2nd with the smallest territorial area. It is important to consider that the response rate of the Portuguese Fab Labs is the highest with 64% of respondents, Italy is the second highest with a response rate just above 24%, and therefore this can show the interest that Portuguese Fab Labs have in this study, maybe also as a yardstick, to compare their reality with the main European and American ones, and to be able to take advantage of this comparison to improve themselves. For the realization of future investigations, ideally the number of Fab Labs in Portugal should be greater, to close the discrepancy with the other countries under analysis. In short, considering the scarcity of studies on this topic, mainly in Portugal, this investigation becomes an important landmark for the literature and practice of Fab Labs.

**Author Contributions:** The authors confirm their contributions to this paper as follows: Conceptualization, S.C. (Sérgio Carqueijó), D.R., J.G. and G.S.; Data curation, S.C. (Sandro Carvalho) and L.B.; Formal analysis, M.D. and G.S.; Investigation, S.C. (Sandro Carvalho), D.R., J.G. and G.S.; Methodology, D.R., L.B. and F.M.; Writing—original draft, S.C. (Sérgio Carqueijó), D.R., J.G. and L.B.; Writing—review & editing, S.C. (Sandro Carvalho), D.R., K.Z. and G.S. All authors have read and agreed to the published version of the manuscript.

**Funding:** This research received no external funding.

**Institutional Review Board Statement:** Not applicable.

**Informed Consent Statement:** Not applicable.

**Data Availability Statement:** Data sharing not applicable.

**Conflicts of Interest:** The authors declare no conflict of interest.

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
