# Peer review of "The Importance of Fab Labs in the Development of New Products toward Mass Customization"

_sustainability, doi:10.3390/su14148671_

Round 1

Reviewer 1 Report

 Too much cited literature. Too much literature is cited throughout the paper ( such as lines 35,78.82,84) where a maximum of literatures are needed. Conclusion are made on a very small number of responses from the laboratory (eg France 151 lab. answers 16; Germany 46 Lab. answers 5) how adequate is the comparison? The conclusion is too large so that the reader does not have a concrete conclusion, it is necessary to express it more clearly.

Author Response

 Too much cited literature. Too much literature is cited throughout the paper ( such as lines 35,78.82,84) where a maximum of literatures are needed.

R- Regarding citations, the text has been improved

 Conclusion are made on a very small number of responses from the laboratory (eg France 151 lab. answers 16; Germany 46 Lab. answers 5) how adequate is the comparison?

R- The comparison of best practices is adequate.

The conclusion is too large so that the reader does not have a concrete conclusion, it is necessary to express it more clearly.

R- The conclusions were rewritten.

Reviewer 2 Report

The article was very well thought out. The authors describe the methodology and literature review well, and the summary and conclusions are well-written. Fab Labs are well discussed and explained. The article, however, has shortcomings that relate to the abstract. The abstract needs to be rewritten. The authors didn't say what the point of the study was or why they chose this topic. They also didn't say anything about the results. I recommend filling these gaps. The work in the introduction lacks information on what gap is filled by the work and what innovative authors bring to the work. I recommend that you complete this point. 

Author Response

The article was very well thought out. The authors describe the methodology and literature review well, and the summary and conclusions are well-written. Fab Labs are well discussed and explained. The article, however, has shortcomings that relate to the abstract.

Thanks

 The abstract needs to be rewritten.

 R_Done

The authors didn't say what the point of the study was or why they chose this topic.

R- In this work it was intended to compare the practice of international Fab Labs with the Portuguese Fab Labs. The conclusion is that the Portuguese Fab Labs need to improve. Improvement solutions are proposed.

 They also didn't say anything about the results.

R- The conclusions were rewritten

 I recommend filling these gaps. The work in the introduction lacks information on what gap is filled by the work and what innovative authors bring to the work.

R- the gaps found are in the conclusions, when we compare the Portuguese Fab Labs with those of other countries. Improvement proposals are made to overcome these gaps

I recommend that you complete this point. 

R-Thanks

Reviewer 3 Report

The network of digital fabrication laboratories is essential to promote innovation, new product development, and support initiatives like Industry 4.0 and the optimization of production lots. This paper studies Fab Labs in the context of Portugal, which is still in the early stages of development.

The study is relevant but there are also opportunities for improvement. The review is presented as follows.

Abstract: The text reveals what was done, items evaluated, and some aspects of differentiation. However, it could be more apparent (1) why this research is relevant and (2) what could be done to improve the scenario of Fab Labs in Portugal.

Introduction

Please avoid references like [1][2][3][4][5][6][7]. It is suggested to include only the most relevant for that sentence/paragraph. A reference should provide additional information specific to the quote, not a suggestion for further reading. Please see other cases.

Page 2, line 66. I suggest separating the presentation of Fab Labs, the paragraph is too long, and it is important to clarify the importance and characteristics of these structures to the reader. The part starting in “Through the case of” is confusing, with very long sentences and many references.

At the end of the introduction is suggested to explain the paper structure.

Section 2.

-Please improve the flow of the first paragraph. The second sentence seems incomplete. 

-Perhaps just ('Google Forms') instead of called 'Google Forms' because it is very popular.

Section 3

-What are control cards?

-Page 7, the paragraph “Regarding security” is not clear

- The single component “Support, training and product realization” seems vague and not distinctive from other types of structures. An interesting characteristic of a Fab Lab is that it provides “digital production infrastructure” / practical development partnerships, not only consulting or training, that are easier to find in the market.

- Table 15 is nor simple to compare only with the count of Fab Labs in each country. Perhaps using the # compared to gross profit, number of companies, area of the country?

- The number of tables in sequence does not help clarify the differences. I suggest including more specific details. It is easy for the reader to get lost in all the differences. For example, why table 28? The number of tables seems excessive.

- Table 23 – Fresadoras CNC, please check the translation

Section 4

- Please rewrite the sentences “They are places where anything can be done” or “by a lack of knowledge at certain levels in 464 Portugal.”

- I was expecting to find a set of recommendations for Portuguese Fab Labs, which is still too dependent on public funds. For example, this type of structure should increase in size and be integrated in more comprehensive networks, or should it focus on specific products? How can the government assist Fab Labs in their cooperation with companies to progressively increase revenues? I am not sure if the best practices in other countries could be transferred to the Portuguese setting or if a specific strategy associated with some economic sectors could be more effective.

In summary, this is very interesting and relevant research. The differentiation is essential to put Fab Labs in a better position, and this paper could provide suggestions.

Author Response

The network of digital fabrication laboratories is essential to promote innovation, new product development, and support initiatives like Industry 4.0 and the optimization of production lots. This paper studies Fab Labs in the context of Portugal, which is still in the early stages of development.

The study is relevant but there are also opportunities for improvement. The review is presented as follows.

Abstract: The text reveals what was done, items evaluated, and some aspects of differentiation. However, it could be more apparent (1) why this research is relevant and (2) what could be done to improve the scenario of Fab Labs in Portugal.

R – Done. The abstract and conclusions was rewritten. New text:

This work is relevant because it compares the Fab Labs of developed countries with those of Portugal. To overcome the difference in good practices existing in other countries, the Portuguese Fab labs need: 1) better publicity, more support for volunteer workers at Fab Labs. Then more ideas will appear and therefore more products; 2) facilitate the use of Fab Labs to have more volunteer workers, who must receive experimental courses, in order to make the best use of the available equipment; 3) evolve, from the current subtractive manufacturing to the additive manufacturing looking for innovation; 4) improve quality, ergonomics and safety in the design of their own products; 5) those responsible should recognize, disseminate and reward the best ideas that have turned into good products, in order to disseminate good practices.

Introduction

Please avoid references like [1][2][3][4][5][6][7]. It is suggested to include only the most relevant for that sentence/paragraph. A reference should provide additional information specific to the quote, not a suggestion for further reading. Please see other cases.

R- Done. This references were reworked. Thanks

Page 2, line 66. I suggest separating the presentation of Fab Labs, the paragraph is too long, and it is important to clarify the importance and characteristics of these structures to the reader.

R- Done. The paragraph was rewritten.

The part starting in “Through the case of” is confusing, with very long sentences and many references.

R-Done. The sentence  was divided and rewritten

At the end of the introduction is suggested to explain the paper structure.

 R- Done. Thanks for suggestion

Section 2.

-Please improve the flow of the first paragraph. The second sentence seems incomplete. 

R- Done. The paragraph was rewritten.

-Perhaps just ('Google Forms') instead of called 'Google Forms' because it is very popular.

 R- Done

Section 3

-What are control cards?

R- Done. Quality control charts. Thanks

-Page 7, the paragraph “Regarding security” is not clear

R_Done. The paragraph was rewritten.

- The single component “Support, training and product realization” seems vague and not distinctive from other types of structures. An interesting characteristic of a Fab Lab is that it provides “digital production infrastructure” / practical development partnerships, not only consulting or training, that are easier to find in the market.

R-done. Support and training is important, because people need to know how to work with machines to make the own product of own design from a good idea.

- Table 15 is nor simple to compare only with the count of Fab Labs in each country. Perhaps using the # compared to gross profit, number of companies, area of the country?

you take an idea and come out with a product made in Fab Labs. How many places in each country can you do it? That's what you wanted to know.

- The number of tables in sequence does not help clarify the differences. I suggest including more specific details. It is easy for the reader to get lost in all the differences. For example, why table 28? The number of tables seems excessive.

R- I think that is correct. Table 28. Kruskal-Wallis test is necessary.

The number of tables does not seem excessive to me. We have to compare and for that we need tables. We compare many countries.

The Kruskal–Wallis test by ranks, or one-way ANOVA on ranks is a non-parametric method for testing whether samples originate from the same distribution. It is used for comparing two or more independent samples of equal or different sample sizes. It is the case.

The Kruskal–Wallis test by ranks,  or one-way ANOVA on ranks is a non-parametric method for testing whether samples originate from the same distribution. It is used for comparing two or more independent samples of equal or different sample sizes. It extends the Mann–Whitney U test, which is used for comparing only two groups. The parametric equivalent of the Kruskal–Wallis test is the one-way analysis of variance (ANOVA).

A significant Kruskal–Wallis test indicates that at least if one sample stochastically dominates one other sample. The test does not identify where this stochastic dominance occurs or for how many pairs of groups stochastic dominance obtains. For analyzing the specific sample pairs for stochastic dominance, Dunn's test, pairwise Mann–Whitney tests with Bonferroni correction, or the more powerful but less well known Conover–Iman test are sometimes used.

Perhaps we need more tests

- Table 23 – Fresadoras CNC, please check the translation

 R-done, tanks

Section 4

- Please rewrite the sentences “They are places where anything can be done” or “by a lack of knowledge at certain levels in 464 Portugal.”

R_done

- I was expecting to find a set of recommendations for Portuguese Fab Labs, which is still too dependent on public funds. For example, this type of structure should increase in size and be integrated in more comprehensive networks, or should it focus on specific products?

R-  Done. The Fab Labs should it focus on ideas that can be transformed into new products

How can the government assist Fab Labs in their cooperation with companies to progressively increase revenues? I am not sure if the best practices in other countries could be transferred to the Portuguese setting or if a specific strategy associated with some economic sectors could be more effective.

R-  Done. Ideas capable of being turned into products are needed. Knowing what other countries are doing will help those who are further behind

In summary, this is very interesting and relevant research. The differentiation is essential to put Fab Labs in a better position, and this paper could provide suggestions.

R- Thanks

Round 2

Reviewer 1 Report

It is necessary to revise the conclusion, to make it shorter and clearer.

Author Response

It is necessary to revise the conclusion, to make it shorter and clearer.

R-  The paper was improved  (in red). The conclusions were rewritten..

Round 3

Reviewer 1 Report

The conclusions were rewritten, it is necessary to revise the conclusion, to make it shorter and clearer.